# A new class of bilayer kagome lattice compounds with Dirac nodal lines and pressure-induced superconductivity

Mengzhu Shi[1,8], Fanghang Yu[1,8], Ye Yang[1], Fanbao Meng[1], Bin Lei [1], Yang Luo[1], Zhe Sun[2], Junfeng He [1], Rui Wang [3], Zhicheng Jiang[4], Zhengtai Liu [4], Dawei Shen [4], Tao Wu [1,5], Zhenyu Wang [1,5], Ziji Xiang [1], Jianjun Ying[1,5 ✉] & Xianhui Chen [1,5,6,7 ✉]

Kagome lattice composed of transition-metal ions provides a great opportunity to explore the intertwining between geometry, electronic orders and band topology. The discovery of multiple competing orders that connect intimately with the underlying topological band structure in nonmagnetic kagome metals $AV_3Sb_5$ ($A$ = K, Rb, Cs) further pushes this topic to the quantum frontier. Here we report a new class of vanadium-based compounds with kagome bilayers, namely $AV_6Sb_6$ ($A$ = K, Rb, Cs) and $V_6Sb_4$, which, together with $AV_3Sb_5$, compose a series of kagome compounds with a generic chemical formula $(A_{m-1}Sb_{2m})(V_3Sb)_n$ ($m$ = 1, 2; $n$ = 1, 2). Theoretical calculations combined with angle-resolved photoemission measurements reveal that these compounds feature Dirac nodal lines in close vicinity to the Fermi level. Pressure-induced superconductivity in $AV_6Sb_6$ further suggests promising emergent phenomena in these materials. The establishment of a new family of layered kagome materials paves the way for designer of fascinating kagome systems with diverse topological nontrivialities and collective ground states.

[1] CAS Key Laboratory of Strongly-coupled Quantum Matter Physics, Department of Physics, University of Science and Technology of China, Hefei, Anhui 230026, China. [2] National Synchrotron Radiation Laboratory, University of Science and Technology of China, Hefei, Anhui 230029, China. [3] Institute for Structure and Function & Department of physics & Center for Quantum Materials and Devices, Chongqing University, Chongqing 400044, China. [4] State Key Laboratory of Functional Materials for Informatics, Shanghai Institute of Microsystem and Information Technology, Chinese Academy of Sciences, Shanghai 200050, China. [5] CAS Center for Excellence in Superconducting Electronics (CENSE), Shanghai 200050, China. [6] CAS Center for Excellence in Quantum Information and Quantum Physics, Hefei, Anhui 230026, China. [7] Collaborative Innovation Center of Advanced Microstructures, Nanjing University, Nanjing 210093, China. [8] These authors contributed equally: Mengzhu Shi, Fanghang Yu. ✉email: yingjj@ustc.edu.cn; chenxh@ustc.edu.cn

The lattice geometry and crystalline symmetry are key factors determining the electronic properties of a crystal. One of the most prominent examples is the kagome lattice, a two-dimensional (2D) corner-sharing triangular network, which recently emerges as a rich frontier for exploring topological and correlated electronic phenomena[1–11]. Owing to its unique lattice geometry, the kagome lattice naturally incorporates linear band crossings hosting Dirac fermions[4,5] as well as destructive interference-derived flat bands in its electronic structure. During the past decade, the topological aspects of kagome lattice have been intensively studied in transition-metal kagome magnets[11–18]; various topologically nontrivial electronic ground states associated with different forms of magnetism have been realized, such as (massive) Dirac fermions and flat bands in ferromagnetic $Fe_3Sn_2$[12,13] and antiferromagnetic FeSn[14], Chern-gapped Dirac fermions in ferromagnetic $TbMn_6Sn_6$[15], Weyl fermions in the ferromagnet $Co_3Sn_2S_2$[16,17] and the noncollinear antiferromagnet $Mn_3Sn$[18], and so on.

More interestingly, in the absence of magnetism, electron correlations could also provoke the emergence of unusual electronic states in kagome lattices[7–10]. A notable example is the recently discovered topological kagome metals $AV_3Sb_5$ ($A$ = K, Rb, Cs)[19,20]. This family of materials not only carries a nontrivial $Z_2$ topological index with band inversion[20], but also hosts cascade of symmetry-breaking electronic orders including (potentially chiral) charge density wave (CDW)[21], nematic/sematic order[22] and superconductivity[20]. Subsequent studies have revealed involved intertwining between these electronic orders that gives rise to numerous exotic phenomena[23], including intrinsic anomalous Hall effect[24,25], unusual competition between CDW and superconductivity[26,27], pair density wave order[28], and possible Majorana zero modes inside the superconducting vortex core[29]. It has been suggested that $AV_3Sb_5$ resembles the high-$T_c$ superconductors in view of the enriched low-temperature ($T$) orderings[23]. Whilst the ongoing investigations provide glimpse to the rich interplay between topology and correlations in kagome lattice, the experimental realization has been still limited, in great part owing to the rarity of kagome materials. Here, we report a new class of vanadium compounds which contains kagome bilayers in their crystal structures. The two members in this class, namely $AV_6Sb_6$ ($A$ = K, Rb, Cs) and $V_6Sb_4$, host different types of Dirac nodal lines in their band structures. Moreover, superconductivity is realized in the former under pressure. These results identify the bilayer vanadium-based kagome compounds as ideal candidates for studying the topological nontriviality and its fingerprints on the electronic properties in kagome lattice.

## Results and discussion

### Crystal structures and physical properties for the bilayer kagome compounds.

In Fig. 1a, b we illustrate the comparison of the crystal structures for the single-layer kagome compounds $AV_3Sb_5$ ($A$ = K, Rb, Cs) and the bilayer kagome compounds: $AV_6Sb_6$ ($A$ = K, Rb, Cs) and $V_6Sb_4$. All these materials consist of layered structural units stacking along the crystallographic $c$ axis. A unit cell of $AV_3Sb_5$ hosts a single $V_3Sb$ layer in which the vanadium sublattice forms a perfect 2D kagome net and the Sb atoms locate at the center of all the kagome hexagons[19] (upper panel in Fig. 1a); such $V_3Sb$ layers are separated by graphite-like $Sb_2$ net and a triangular sublattice of alkaline cations $A$ (left panel in Fig. 1b). The resulted hexagonal structure corresponds to the space group $P6/mmm$. By contrast, the bilayer compounds $AV_6Sb_6$ and $V_6Sb_4$ crystalize in the rhombohedral space group $R\bar{3}m$ (No. 166); in their unit cells two adjacent $V_3Sb$ kagome slabs with in-plane offsets of V sites[30] form a structural unit (lower panel in Fig. 1a). In $AV_6Sb_6$ these $(V_3Sb)_2$ bilayers are sandwiched

by $Sb_2$ sheets and triangular $A$ sublattices that are the same as those in $AV_3Sb_5$ (Fig. 1b), whilst in $V_6Sb_4$ the intercalated $A$ cations are absent and there is a single $Sb_2$ net between two $(V_3Sb)_2$ bilayers. We note that the structure of $V_6Sb_4$ is identical to the ferromagnetic kagome compound $Fe_3Sn_2$[12,30], whereas the $AV_6Sb_6$ series adopts an unprecedented crystal structure of kagome compounds (for detailed structural parameters, see Supplementary Tables 1 and 2). Although the single-layer and bilayer compounds have different crystalline symmetries, we stress that they can be represented by a generic chemical formula $(A_{m-1}Sb_{2m})(V_3Sb)_n$ ($m$ = 1, 2; $n$ = 1, 2) with the values of ($m$, $n$) for $AV_3Sb_5$, $AV_6Sb_6$ and $V_6Sb_4$ being (2, 1), (2, 2) and (1, 2), respectively. Hence, the crystal structure of the entire series $(A_{m-1}Sb_{2m})(V_3Sb)_n$ can be viewed as an alternate stacking of the $(A_{m-1}Sb_{2m})$ blocks and the $(V_3Sb)_n$ blocks.

Single crystals of $AV_6Sb_6$ ($A$ = K, Rb, Cs) and $V_6Sb_4$ were synthesized using a self-flux growth technique (Methods). The high quality of these single crystals is confirmed by our X-ray diffraction (XRD) measurements: the reconstructed (hk0) plane determined from single-crystal XRD displays sharp spots with sixfold rotational symmetry (Fig. 1d, e) and a $2\theta$-$\omega$ scan yields only a series of sharp (00l) Bragg peaks (Fig. 1c). We summarize the results of the single-crystal XRD analysis for $V_6Sb_4$, $RbV_6Sb_6$, and $CsV_6Sb_6$ in Supplementary Tables 1 and 2. Electrical transport and magnetization measurements identify all the bilayer compounds as nonmagnetic metals without indications of additional orderings (Fig. 1f and Supplementary Fig. S1). We note that no CDW ordering is observed in our bilayer kagome samples, contrary to the case in the single-layer $AV_3Sb_5$ ($A$ = K, Rb, Cs) where a CDW transition occurs at 80–100 K[19]. Hall resistivity measured in $CsV_6Sb_6$ (inset of Fig. 1f) indicates dominant electron-type carriers with the density $n_e = 2.1 \times 10^{21}$ cm$^{-3}$.

### First-principles calculations on $CsV_6Sb_6$ and $V_6Sb_4$.

To further investigate the electronic structures of those bilayer kagome compounds, we performed first-principles calculations based on the density functional theory (DFT). The calculation results approve that $AV_6Sb_6$ ($A$ = K, Rb, Cs) and $V_6Sb_4$ lack local magnetic interactions and possess nonmagnetic ground states, consistent with the experimental observations (Supplementary Fig. 1). Here, we mainly focus on the compounds $CsV_6Sb_6$ and $V_6Sb_4$. The calculated band structures for the other two members in the isostructural series $AV_6Sb_6$ with $A$ = K, Rb are presented in Supplementary Fig. 2. In Fig. 2a, b we show the band structures as well as the projected density of states (DOS) for $CsV_6Sb_6$ and $V_6Sb_4$, respectively. The corresponding high-symmetry paths of the BZ of a rhombohedral lattice are depicted in Fig. 2c. The DOS in the vicinity of the Fermi level are dominated by V-3d orbitals for both $CsV_6Sb_6$ and $V_6Sb_4$, whereas the Sb-p orbitals have rather weak contributions (Fig. 2a, b). Remarkably, the calculated band structures of both compounds exhibit linear band crossings (i.e, Dirac points) that are close to the Fermi level along the specific high-symmetry paths, Γ-Z-F-Γ-L-Z-P (Fig. 2a, b). To be mentioned, the electronic structure of the bilayer kagome compounds is distinct from that of the single-layer $AV_3Sb_5$ which are hallmarked by multiple Dirac crossings and saddle points near the Fermi level[20,31].

Our DFT calculations reveal that $CsV_6Sb_6$ is a Dirac nodal line semimetal. As shown in Fig. 2a, the band crossings occur along the high-symmetry paths from Γ/Z to the high-symmetry points at the boundaries of the Brillouin zone (BZ) such as the P, F, and L. All these high-symmetry paths lie in a middle plane of the BZ (highlighted in yellow in Fig. 2c); on this plane, the band crossings form two type-II Dirac nodal lines (shown as the red lines in Fig. 2c) characterized by tilted Dirac cones[32]. By further checking the little group of the BZ, we find that these type-II Dirac nodal lines are

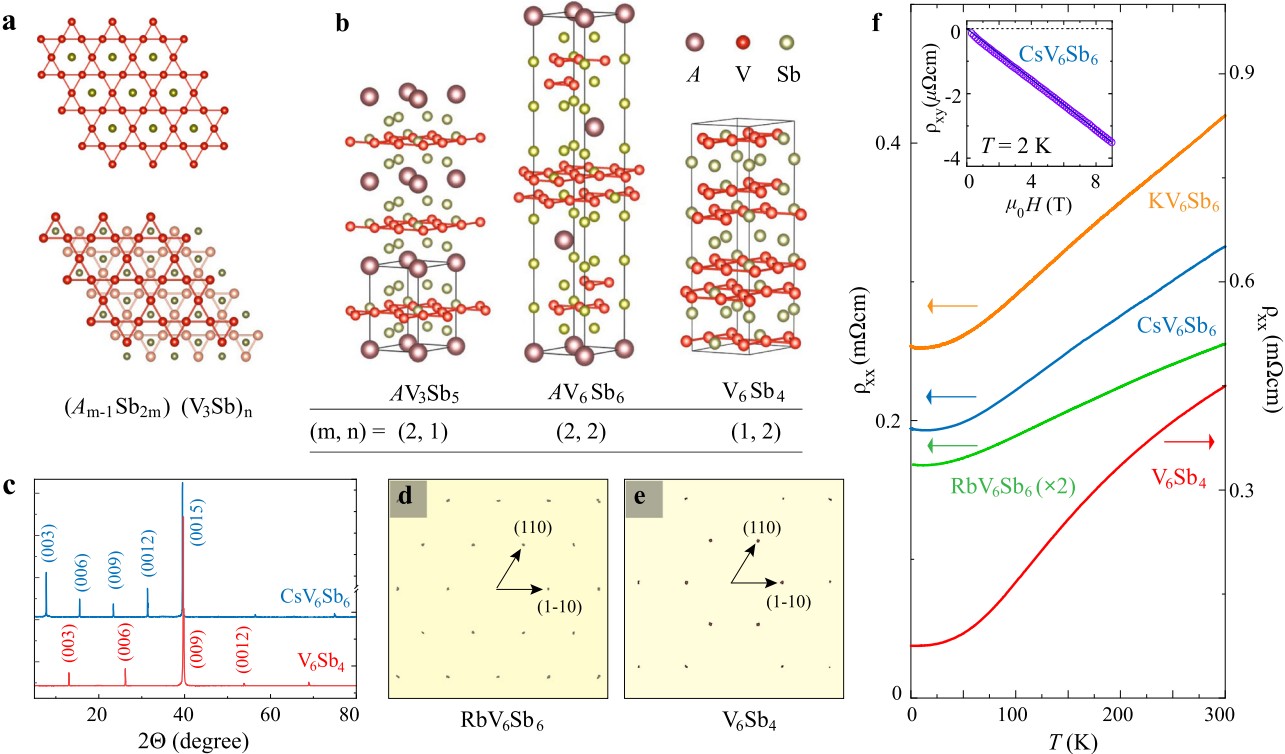

**Fig. 1 Structural and transport properties of the bilayer vanadium-based kagome compounds. a** Top view of (top) the $V_3Sb$ kagome nets in single-layer compounds $AV_3Sb_5$ ($A$ = K, Rb, Cs), (bottom) $(V_3Sb)_2$ bilayers in $AV_6Sb_6$ ($A$ = K, Rb, Cs) and $V_6Sb_4$. The different offsets of the two adjacent $V_3Sb$ layers lower the rotational symmetry from sixfold to threefold. **b** Sketches of the structural unit cells of (left) $AV_3Sb_5$, (middle) $AV_6Sb_6$, (right) $V_6Sb_4$. The structures of all three series can be described as alternate stacking of the kagome unit $(V_3Sb)_n$ and the spacing unit $A_{m-1}Sb_{2m}$. Single-crystal X-ray diffraction pattern for **c** the (00$l$) direction and **d**, **e**, the ($hk0$) plane measured in bilayer kagome compounds. **f** Temperature-dependent electrical resistivity of $KV_6Sb_6$ (orange), $RbV_6Sb_6$ (green, amplified by a factor of 2), $CsV_6Sb_6$ (blue), and $V_6Sb_4$ (red). Inset shows the Hall resistivity measured in $CsV_6Sb_6$ at $T$ = 2 K. The solid dark blue curve is the linear fit corresponding to an electron-type carrier density of $n_e = 2.1 \times 10^{21}$ cm$^{-3}$.

symmetry protected because the two intersecting bands along the above arbitrary high-symmetry paths belong to different irreducible representations (IR) $\Gamma_1$ and $\Gamma_2$ of the mirror symmetry $C_s$ (see the insets in Fig. 2a). Due to the three-fold rotational symmetry, there are three equivalent middle planes, and thus six type-II Dirac nodal lines that are symmetrically distributed in the BZ of $CsV_6Sb_6$. On the other hand, the crossings of valence and conduction bands are absent along $\Gamma/Z$-L in the band structure of $V_6Sb_4$ (Fig. 2b). However, the preserved band crossings along $\Gamma/Z$-F can still form symmetry-protected nodal lines on three equivalent middle planes. Contrary to the type-II nodal lines in $CsV_6Sb_6$, these nodal lines in $V_6Sb_4$ are type-I and feature closed loops around the band inverted point F (Supplementary Fig. 2).

In the presence of the spin-orbit coupling (SOC), the spin-rotation symmetry is broken, subsequently, the nodal lines in a system with the coexistence of spatial inversion and time-reversal symmetries are always destroyed[33]. As shown in the insets of Fig. 2a, b, when the SOC effect is included, the two intersecting bands without SOC now belong to the same IR $\Gamma_4$ of the mirror symmetry $C_s$. Thus the band crossings are avoided. Nonetheless, the SOC effect is rather weak in the vanadium bilayer kagome compounds: the gap opened at the Dirac band crossings is almost negligible; in particular, for $CsV_6Sb_6$ the gap width is less than 1 meV. Therefore, its type-II Dirac nodal lines are nearly intact. With the existence of the SOC gaps, we can use parity products of occupied bands at the time-reversal invariant momenta (TRIM) points to reveal the topological nontriviality[34]. For a nonmagnetic compound crystalizing in the $R\bar{3}m$ space group, the three F (and the three L) points are equivalent, thus we only calculate the parity eigenvalues at four TRIM points, i.e., the $\Gamma$, Z, F, and L

points (for detailed calculation results, see Supplementary Table 3). As illustrated in Fig. 2d, the parity analysis indicates that bands 115 and 117 (95 and 99) are topologically nontrivial in $CsV_6Sb_6$ ($V_6Sb_4$). These nontrivial bands are close to the Fermi level and their cooperation with nodal fermions would be expected to generate rich exotic quantum phenomena. It should be noticed that the inclusion of Hubbard $U$ in the DFT calculations does not change the band topology (Supplementary Fig. 3), and the Dirac nodal lines remain intact with $U$ = 2.0 eV.

**ARPES measurements on $CsV_6Sb_6$.** The band structure calculation results for $CsV_6Sb_6$ are supported by our angle-resolved photoemission spectroscopy (ARPES) measurements. As shown in Fig. 3a–f, the constant energy contours at binding energies $E_b$ = 0, 200 and 400 meV, as measured with 60 eV photons, are similar to those of the DFT+$U$ calculations ($U$ = 2eV). At the Fermi energy, we observe relatively high intensities close to the boundary of the projected 2D BZ (marked by yellow lines in Fig. 3b) and a hexagonal contour around the $\bar{\Gamma}$ point; both of which are consistent with the DFT results (Fig. 3a). At $E_b$ = 200 meV, the most prominent features of the calculated constant energy contours are the hexagonal pocket around the $\bar{\Gamma}$ point and rounded-triangular pockets centered at the $\bar{K}$ points (Fig. 3c), whereas a gear-shaped pocket centered at the $\bar{\Gamma}$ develops at higher binding energies (Fig. 3e). These features are well reproduced in the photoemission intensity maps (Fig. 3d, f). Figure 3g shows the calculated bulk band dispersion along the $\bar{\Gamma}$-$\bar{M}$ direction (red dashed line in Fig. 3b) with different $k_z$ ranging from 0 to $2\pi/c$, and one can find that the $k_z$ dependence of the bulk bands in this

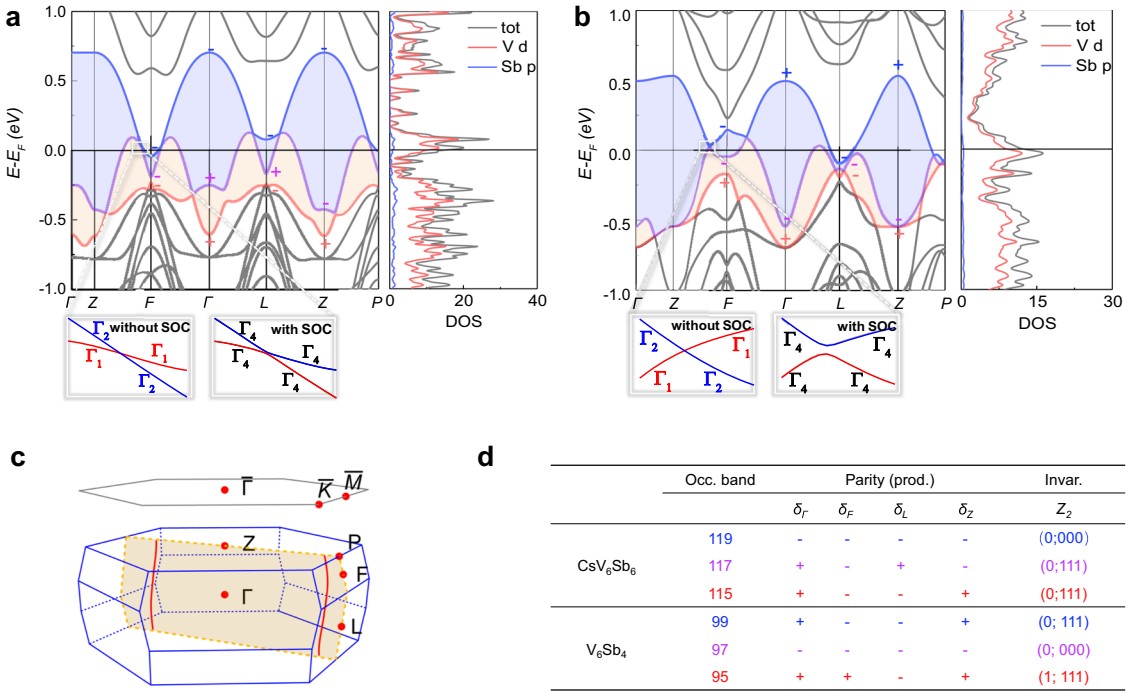

**Fig. 2 Dirac nodal rings in the electronic structure of $CsV_6Sb_6$ and $V_6Sb_4$.** The band structures and projected density of states (DOS) were obtained from DFT calculations for **a** $CsV_6Sb_6$ and **b** $V_6Sb_4$. The blue and yellow shaded areas highlight the topologically nontrivial bands near the Fermi level and the band gaps between them. The calculated parities are also shown for these bands at the TRIMs. The insets of panels **a**, **b** are zoom-ins of the Dirac band crossings along the Z-F direction without (left) and with (right) spin-orbit couplings. **c** The surface and bulk Brillouin zones of the rhombohedral $AV_6Sb_6$ and $V_6Sb_4$. The cross-section marked by yellow color is the middle plane in which the high-symmetry paths are selected for the DFT calculations in **a**, **b**. The two red curves denote the position of type-II Dirac nodal lines in $CsV_6Sb_6$. **d** The parity at TRIMs and the $Z_2$ invariant for each band close to the Fermi level in $CsV_6Sb_6$ and $V_6Sb_4$.

material is very weak, which is also confirmed by the photon-energy-dependent ARPES measurements (Supplementary Figs. 4, 5). For a direct comparison, in Fig. 3h we display the corresponding ARPES-intensity plot. An overall agreement between the ARPES data and the calculated bulk bands (Fig. 3g). In particular, the two crossing bands (bands 117 and 119 in Fig. 2a) contributing to the type-II Dirac nodal lines in the $\bar{\Gamma}$-$\bar{M}$ direction are resolved, with the Dirac crossings located in close vicinity to the $E_F$. We note that the band structure of $CsV_6Sb_6$ is significantly different from its single-layer counterpart $CsV_3Sb_5$[13,31], indicating the importance of inter-$(V_3Sb)$-layer coupling in these kagome lattice compounds.

**Pressure-induced superconductivity in $AV_3Sb_5$.** Despite that the bilayer vanadium-based kagome compounds, unlike $AV_3Sb_5$ $(A = K, Rb, Cs)$[20,26], do not exhibit superconductivity at ambient pressure, we realize superconductivity in $AV_6Sb_6$ $(A = K, Rb, Cs)$ by applying quasi-hydrostatic pressures. The results of high-pressure resistance measurements on two $CsV_6Sb_6$ samples are presented in Fig. 4a. The $T$-dependences of the normalized resistance $R/R_{300K}$ show metallic behavior in the entire pressure range with the residual resistivity ratio (RRR) gradually decreases below 15 GPa. At 15.8 GPa it drops to ~1.2. With pressure further increasing, superconducting transition emerges at 21.2 GPa manifested by a pronounced resistance drop (Fig. 4b). Magnetic fields can gradually suppress the transition temperature, which confirms that the resistance drop is due to a superconducting transition (Supplementary Fig. 6). The transition temperature determined by $T_c^{90\%}$ (i.e., where the resistance drop to 90% of the normal state value) is 1.04 K. The evolution of $T_c^{90\%}$ under applied pressure is nonmonotonic: it first increases rapidly and reaches a maximum of

1.48 K at 33.0 GPa, then starts to decrease slowly; superconductivity persists up to the highest pressure (79.5 GPa) achieved in this measurement, where $T_c^{90\%}$ is ~1 K (Fig. 4b). Such nonmonotonic behavior gives rise to a broad dome-shaped superconducting regime in the high-pressure phase diagram (Fig. 4c), suggesting a complex interplay of the pressure-dependent DOS and structural instabilities[35,36]. Similar superconducting dome is also observed in $RbV_6Sb_6$ and $KV_6Sb_6$ under high pressure, yet with lower maximum $T_c$ (Supplementary Fig. 7). Intriguingly, superconductivity in all three materials appears in the vicinity of the minimum of RRR (Fig. 4c and Supplementary Fig. 7). The correlation between $T_c$ and RRR indicates that the emergence of superconductivity is associated with electronic structure modifications.

We note that such modification stems from a structural phase transition that occurs above ~20 GPa, where the structure changes from rhombohedral to monoclinic as revealed by our high-pressure XRD measurements (Supplementary Figs. 8, 9). It is most likely that the monoclinic phase hosts superconductivity. Refinements of the XRD data provide further details of the structural transition: the ambient-pressure rhombohedral phase evolves into the monoclinic phase via a lattice distortion at which the in-plane lattice parameter $a$ develops into two unequal values $a$ and $c$, whilst the angle $\beta$ changes from 120° to ~110° (Supplementary Fig. 8). The pressure-induced superconductivity accompanied by a structural transition in $AV_6Sb_6$ resembles that observed in numerous topological semimetals[37–41]. Most notably, in some of these materials, the superconductivity is proposed to emerge from topological electronic bands[37,39,40], offering a good opportunity to probe the feasible realization of topological superconductivity. Future studies are needed to determine the topological properties of the high-pressure monoclinic phase and to clarify how superconductivity develops in this phase with lower symmetry.

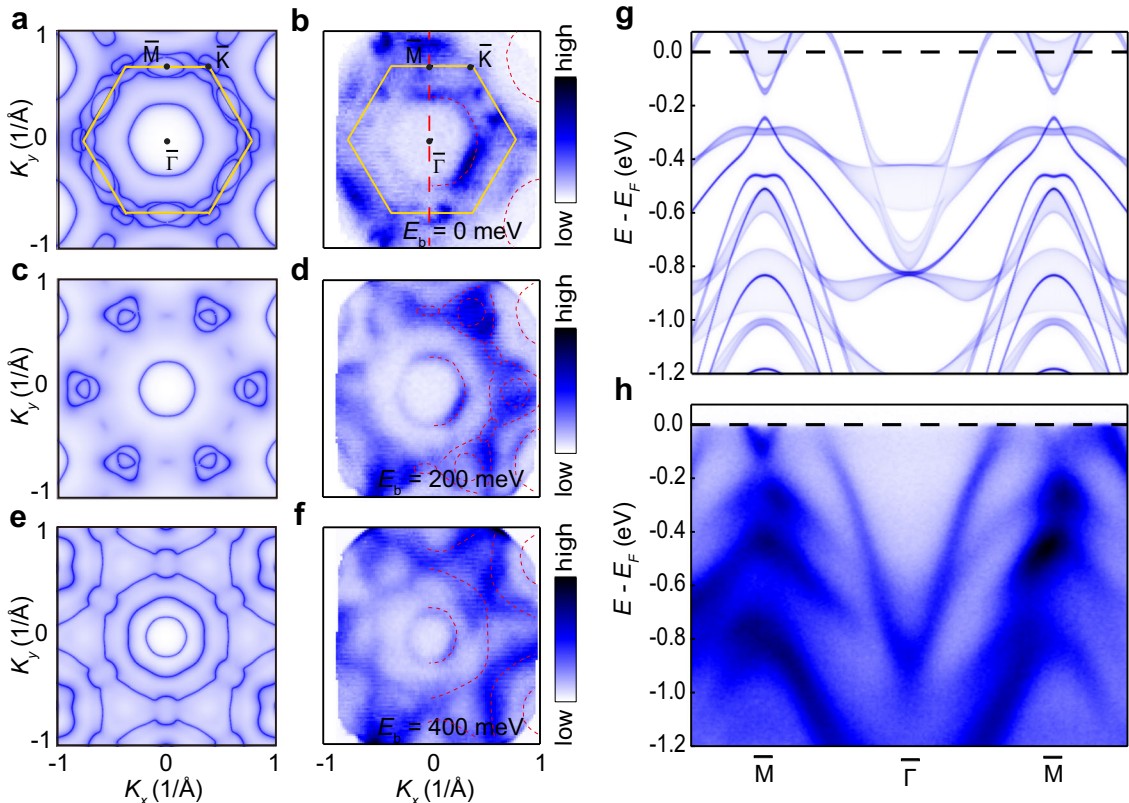

**Fig. 3 Experimentally resolved electronic structure. a** DFT + U-calculated Fermi surface contours at $k_z = 0.5$ with $U = 2$ eV for $CsV_6Sb_6$. **b** Constant energy contours at binding energy $E_b = 0$ measured in $CsV_6Sb_6$ at around 12 K by ARPES using 60 eV photons. Yellow lines denote the boundary of the projected 2D Brillouin zone. **c**, **d**, **e**, **f**, Same as **a**, **b**, but for the constant energy contour at $E_b = 200$ meV and $E_b = 400$ meV. respectively. Red dotted lines in **b**, **d**, **f** are guides to the eye. **g** The bulk band structures of $CsV_6Sb_6$ at different $k_z$ ranging from 0 to 0.5 were obtained from DFT+U calculations with $U = 2$ eV. **h** Photoelectron intensity plot along the $\bar{\Gamma} - \bar{M}$ momentum cut (red dashed line in b), measured with 60 eV photons at $T \approx 12$ K.

Our theoretical and experimental investigations reveal unusual topological metal phases in the bilayer vanadium-based kagome compounds. Distinct from the typical kagome physics concerning Dirac points and van Hove singularity in single-layered $AV_3Sb_5$[20,23], the most important characteristics of the bilayer compounds are denoted by the Dirac nodal lines near the Fermi level. The pressure-induced superconductivity discovered in $AV_6Sb_6$ further suggests promising emergent phenomena in the bilayer kagome materials. All these results provide inspiring perspectives for future explorations of the enriched topological physics in kagome lattice.

## Methods

**Single crystal growth and characterization.** Single crystals of $AV_6Sb_6$ ($A = K$, Rb, Cs) and $V_6Sb_4$ were grown using a self-flux method. For $AV_6Sb_6$, K/Rb/Cs (ingot or liquid 99.5%), V (powder, 99.5%), Sb (shot, 99.999%) were loaded into an alumina container with the molar ratio of 2:3:6 and then sealed into a double-wall silica tube. Mixture was subsequently heated to 1150 °C and kept for 24 h, then slowly cooled to 1050 °C for 72 h. The excess flux was removed by centrifuging at 1050 °C. The products were single crystals with sizes of a few millimeters in *ab* plane and less than 100 μm in thickness. For $V_6Sb_4$, K (ingot, 99.5%), V (powder, 99.5%), Sb (shot, 99.999%) were mixed with the molar ratio of 2:3:6 and sealed in the same double-wall silica tube. The tube was heated to 1150 °C and kept for 24 h, then slowly cooled to 1050 °C in 72 h. After soaked at 1050 °C for 5 h, the temperature was increased to 1100 °C in 20 h. Large crystals of $V_6Sb_4$ with sizes of several millimeters were obtained from the flux by centrifuging at 1100 °C.

Single-crystal X-ray diffraction measurements were carried out on a XtaLAB AFC12 (RINC): Kappa single diffractometer (Rigaku, Japan) with a charge-coupled device detector and Cu source in Core Facility Center for Life Sciences, USTC. The data was processed and reduced using CrysAlisPro[42]. Using Olex-2[43], the structure was solved with the ShelXT structure solution program[44] via direct methods and refined with the ShelXL refinement package[45]. Magnetization measurements were performed on a Quantum Design Magnetic Properties Measurement Systems (MPMS-5). Plate-shaped single crystals were attached to a quartz rod with the magnetic field applied parallel to and perpendicular to the c-axis. The transport

properties were measured on the Quantum Design Physical Properties Measurement System (PPMS-9) using a standard six-probe configuration.

**First-principles calculations.** To depict electronic properties of these novel kagome compounds of $AV_6Sb_6$ and $V_6Sb_4$, we carried out first-principles calculations based on the density functional theory[46] as implemented in the Vienna ab initio simulation package[47]. The exchange-correlation functional was described by generalized gradient approximation with Perdew–Burke–Ernzerhof formalism[48]. The core-valence interactions were treated by projector augmented-wave potentials[49] with a plane-wave-basis cutoff of 450 eV. The Brillouin zone (BZ) was sampled by a $12 \times 12 \times 12$ Monkhorst-Pack grid[50] to simulate the rhombohedral structure. The crystal structures were fully relaxed by minimizing the forces on each atom smaller than $1.0 \times 10^{-3}$ eV/Å, and the van der Waals interactions along the c-layer stacking direction were considered by the Crimme (DFT-D3) method[51]. The topological class was characterized by the $Z_2$ invariants, which are calculated from the parity eigenvalues at TRIM points using IRVSP package[52]. We also employed the DFT + U method to calculate the band structure of $CsV_6Sb_6$, in which the Hubbard U correction represents the on-site Coulomb interactions on the d-orbital of vanadium; the band topology persists upon varying U values (Supplementary Information Fig. S3).

**ARPES measurements.** ARPES measurements were performed at the beamline 13 U of the National Synchrotron Radiation Laboratory at University of Science and Technology of China (photon energy 35 eV), and the beamline 03U of the Shanghai Synchrotron Radiation Facility (SSRF) (photon energy 42–60 eV). The samples were cleaved in situ with a base pressure less than $6 \times 10^{-11}$ torr. We note that the terraced surfaces of the cleaved samples usually hamper a clear observation of the fine electronic features. Further ARPES investigations with improved resolution are required to reveal the possible topological surface states[53] in the $AV_6Sb_6$ materials.

**High-pressure transport measurements.** Diamond anvils with various culets (200–300 μm) were used for high-pressure transport measurements. NaCl was used as a pressure transmitting medium and the pressure was calibrated by using the shift of ruby fluorescence and diamond anvil Raman at room temperature. For

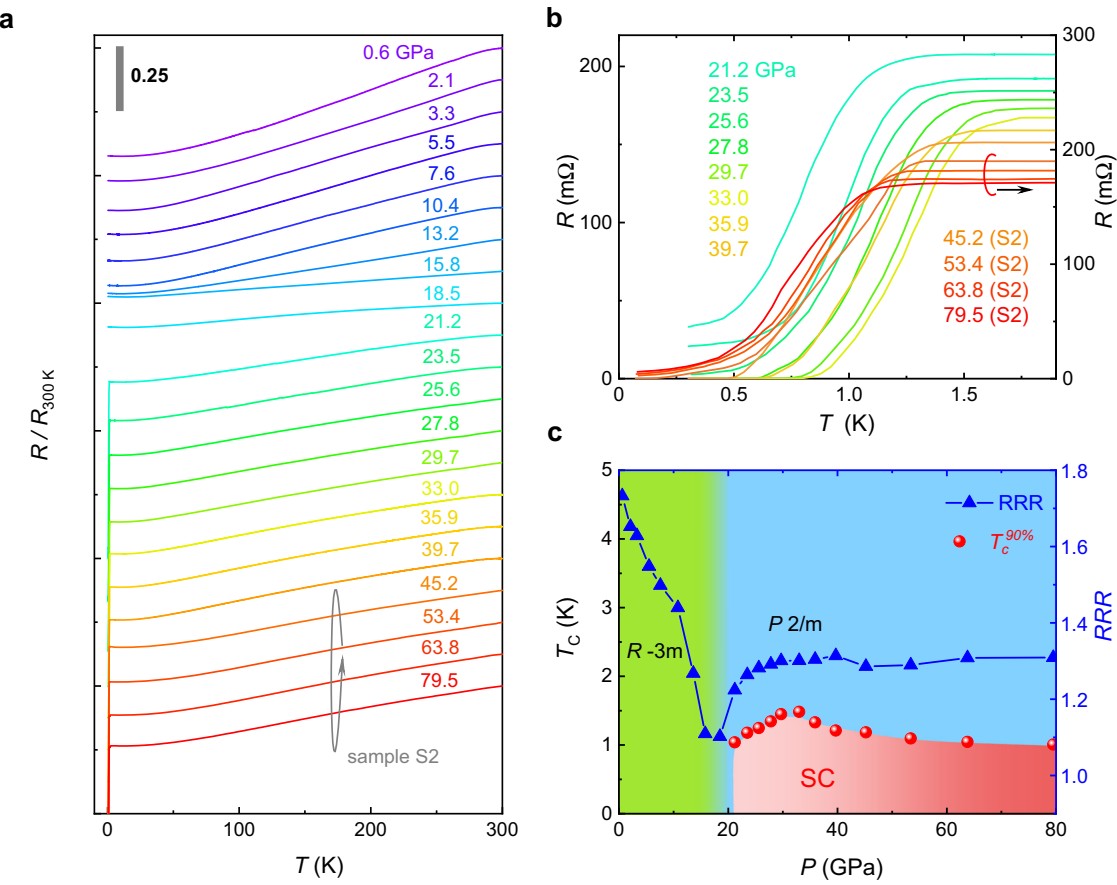

**Fig. 4 Pressure-induced superconductivity in CsV₆Sb₆. a** Temperature dependence of the resistance of $CsV_6Sb_6$ normalized using the room temperature (300 K) value under various pressures up to 79.5 GPa. The curves taken at $P < 40$ GPa and $P > 40$ GPa are measured in samples S1 and S2, respectively. Data are shifted vertically for clearance. The gray vertical bar denotes a scale of 0.25. **b** An expanded view of the low-temperature resistance of $CsV_6Sb_6$ in a pressure range of 21.2–79.5 GPa, showing the superconducting transitions. **c** Phase diagram for $CsV_6Sb_6$ under pressure. The superconducting transition temperature $T_c$ (red solid circles) is determined as the temperature where the resistance drops to 90% of the normal state value. The shaded regimes shown in green and blue represents the rhombohedral (space group $R\bar{3}m$) and monoclinic (space group $P2/m$) structural phases, respectively (see Supplementary Figs. 8, 9).

each measurement cycle, the pressure was applied at room temperature using the miniature diamond anvil cell. The transport measurements were performed in a dilution refrigerator (Kelvinox JT, Oxford Instruments) or a ³He cryostat (HelioxVT, Oxford Instruments). Single-crystalline samples of $AV_6Sb_6$ and $V_6Sb_4$ were cut into typical dimensions of $50 \times 50 \times 10$ μm³. The resistivity was measured using a four-probe configuration. $V_6Sb_4$ does not show indication of superconducting transition up to the highest pressure we achieve, i.e., $P \approx 80$ GPa (Supplementary Fig. 7).

**High-pressure X-ray diffraction.** The high-pressure synchrotron XRD measurements were performed at room temperature at the beamline BL15U1 of SSRF with a wavelength of $\lambda = 0.6199$ Å. A symmetric diamond anvil cell with a pair of 200 μm culet size anvils was used to generate pressure. 70 μm sample chamber is drilled from the Re gasket and Daphne 7373 oil was loaded as a pressure transmitting medium.

## Data availability
All data supporting the findings of this study are available from the corresponding authors upon reasonable request.

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

## Acknowledgements

We thank Shengtao Cui and Soohyun Cho for their assistance in synchrotron ARPES measurements. This work is supported by the National Key Research and Development Program of the Ministry of Science and Technology of China (2017YFA0303001, 2019YFA0704901, and 2016YFA0300201), the Anhui Initiative in Quantum Information Technologies (AHY160000), the Strategic Priority Research Program of the Chinese Academy of Sciences (XDB25000000), the National Natural Science Foundation of China (NSFC, Grants No. 11888101, 11974062, and U20322), the Science Challenge Project of China (TZ2016004) and the Key Research Program of Frontier Sciences, CAS, China (QYZDY-SSW-SLH021). The DFT calculations in this work are supported by the Supercomputing Center of University of Science and Technology of China. High-pressure synchrotron XRD work was performed at the BL15U1 beamline, SSRF in China. Part of this research used Beamline 03U of the Shanghai Synchrotron Radiation Facility, which is supported by ME[2] project under contract No. 11227902 from the National Natural Science Foundation of China.

## Author contributions

X.H.C. conceived the project and supervised the overall research. M.S., F.Y., and J.Y. grew the single crystal samples. M. S. performed the XRD, electrical transport, and magnetization measurements with the help of B.L., and analyzed the data with T.W., Z.W., and Z.X. F.M., Y.L., Z.S., Z.J., Z.L., D.S., and J.H. performed the ARPES experiments and analyzed the resultant data. F. Y and J. Y. performed the high-pressure measurements. Y.Y. and R.W. performed the DFT calculations. Z.W., Z.X., and X.H.C. wrote the manuscript with input from all authors.

## Competing interests

The authors declare no competing interests.
