## [Peer Review File · Nature Communications]

A new class of bilayer kagome lattice compounds with Dirac nodal lines and pressure-induced superconductivityREVIEWER COMMENTS

Reviewer #1 (Remarks to the Author):

In this manuscript, the authors describe the discovery and characterization of new compounds that share similar building motifs with AV3Sb5. AV3Sb5 has been heavily investigated recently as a Kagome lattice that show superconductivity. The newly discovered compounds in this manuscript would offer a new chemical tuning parameter into the physics of interest. Therefore, I think this manuscript presents new information for the community. However, there are two major points that I think need to be addressed.

1. The structural refinement result is questionable. The refinement of V6Sb4, for example, has a very large diff. peak/hole charge number that leads to a very large R value at the end. This likely indicates the structural model does not match with the measured diffraction data. Too much charge density is not accounted for, or over calculated. Similarly, the R value for RbV6Sb6 is not great. On the fundamental side, it is unclear if the actual structure is indeed related to AV3Sb5 since the claimed crystal structure could be wrong.

2. In terms of superconductivity, the transport data obtained from 4 probe measurement does not go to zero resistivity at the pressure the authors claim. In fact, the resistivity at the base temperature is likely still numerically larger than a good metal, say copper, at room temperature. Given that a filamentary superconductor would be able to short a sample already, the bulk nature of superconductivity is highly questionable. Possibility of a structural or magnetic phase transition under high pressure cannot be ruled out.

Reviewer #2 (Remarks to the Author):

In the manuscript entitled «A new class of bilayer kagome lattice compounds with Dirac nodal lines and pressure-induced superconductivity », M. Shi, F. Yu and coworkers report the synthesis of new compounds [AV6Sb6, A = K, Rb, Cs, and V6Sb4] that host kagome bilayers with linearly dispersing bands forming nodal Dirac lines.

Nowadays, the study of Kagome lattices, in which magnetic frustration, topological ordering, superconductivity and charge ordering are simultaneously present, it represents the frontier in the study of topological materials, being AV3Sb5 the main focus of the community. In this respect, the synthesis and study of new compounds sharing similar properties is timely. However, the materials investigated in this manuscript do not exhibit several of the intriguing phenomena that are fascinating the community. As the authors clarify, in fact, those materials have no charge ordering and DFT calculations indicate that they lack magnetic interactions, and a nonmagnetic ground-state is expected.

In general, I found the manuscript well written, and I appreciate the nice description of the symmetry protecting the band degeneracy responsible for the Dirac nodal line. However, in my opinion this is not sufficient to guarantee publication in Nature Communications.

1) The quality of the ARPES data is in fact not sufficient to confirm the theoretical prediction of DFT. The authors themselves are very honest when in the methods they state “the terraced surfaces of the cleaved samples hamper a clear observation of fine electronic features”.

- In particular, we cannot resolve the two states that should be responsible for the Dirac line.

- The assignment of the Fermi level is unclear to me, in fact there are several ARPES signatures that are far from being reproduced by DFT. In Figure 3 e, the high intensity at the M point is clearly within the calculated red lines. Similarly, in Figure 3 b the large intensity coincides with the Brillouin zone boundary indicated by yellow lines, and not with the calculations shown by the red dashed line.

2) Given the symmetry of the crystal structure, I would expect multiple surface terminations to be possible, and the authors should characterize their influence on the electronic properties both from the experimental and theoretical point of view.

3) ARPES data might be dominated by surface states, and photon energy dependent study would be necessary to distinguish the surface from bulk contributions.

4) From the theoretical point of view, I do not fully understand Figure 2 d. Parity is evaluated only at 4 TRIM points, while I would have expected 8 contributions (if symmetry consideration lowers the number of points, it should be explained). Since the materials have not a direct gap, the authors should indicate in Panel a and b, the lines where the parities product has been calculated.

In figure 2 d, why the parities of only few selected bands are shown?

In figure 2 a and 2 b, what do the blue and yellow dashed areas indicate?

5) The report of a superconducting state under pressure is certainly of interest. However, without a proper characterization of the crystal structure/symmetry is impossible to be sure that this is a property of the crystals described in Figure 1-3. As the authors carefully explained, the Dirac nodal line as well as the Kagome lattices arise from the space group and the lattice symmetry. The application of pressure might be responsible for a structural phase transitions, and this is a central aspect to be verified.

6) Finally, although novelty is not the reason why I cannot recommend the publication of this manuscript in Nature Communication, I would like to bring to the attention of the authors a manuscript, recently published in Chinese Physical Letters 38 127401 [Structures and Physical Properties of V-Based Kagome Metals CsV₆Sb₆ and CsV₈Sb₁₂], in which the synthesis of AV₆Sb₆ was reported. Hence, I would avoid the use of the term “discovery” in the abstract.

7) Minor comments:

- The data of Figure 3 b and c are symmetrized. The authors should clearly state this in the caption. If possible add the raw data to the supplementary information.
- In few points of the manuscript the authors indicate the References as “refs N,M”. I do not know if that is an error or it is done by purpose.
- At the end of the first paragraph of page 4 the authors say “along the specific high-symmetry paths”, but it is unclear which paths.
- In general, I do not understand why the Authors stress the possibility of using one single generic stoichiometric formula to describe very different materials, since the authors themselves have nicely discussed how symmetry is the key factor, and not stoichiometry.
- In general DFT is not a good level of approximation to discuss energy band-gap of few meV. I think that besides better ARPES data, the manuscript would profit of refined calculations (hybrid functional for example) to properly check the band dispersion of the states forming the band-gap.

Reviewer #3 (Remarks to the Author):

Shi et al reported a new family of Kagome materials that exhibit topological band structure and superconductivity. Recently, Kagome materials like CsV_3Sb_5 attract tremendous attention for unconventional superconductivity, loop current CDW, and topological states. The present work brings a new group of kagome materials to this family, but with enough differences such as the strong interlayer coupling, nodal line states, and superconductivity under pressure. This work has the potential to represent new features of kagome materials. I would consider it positively for Nature Comm if they can provide more solid information to support their conclusions.

1. The nodal lines and Z2 topological invariants are highlighted in the paper, which is mainly supported by calculations. If the theory is correct, there should be signatures of topological surface states due to the nodal lines in the bulk. The authors should present direct calculations on the surface band structure and also reveal them in ARPES.
2. The materials are heavily electron-doped. In contrast, CVS is commonly hole-doped. Can authors elaborate on the material/chemistry origin behind different doping behaviors?
3. The pressure induces superconductivity. Do authors have any conclusions on the nature of superconductivity? Further, the pressure is rather high up to 80 GPa. Is there any structural transition under pressure? Does the material still preserve the kagome lattice or the topology?

REVIEWER COMMENTS

Reviewer #1 (Remarks to the Author):

In this manuscript, the authors describe the discovery and characterization of new compounds that share similar building motifs with AV₃Sb₅. AV₃Sb₅ has been heavily investigated recently as a Kagome lattice that show superconductivity. The newly discovered compounds in this manuscript would offer a new chemical tuning parameter into the physics of interest. Therefore, I think this manuscript presents new information for the community. However, there are two major points that I think need to be addressed.

We are grateful to Reviewer#1 for confirming the novelty and significance of our work. The Reviewer's valuable comments have also helped us to clarify our findings in the revised manuscript.

1. The structural refinement result is questionable. The refinement of V₆Sb₄, for example, has a very large diff. peak/hole charge number that leads to a very large R value at the end. This likely indicates the structural model does not match with the measured diffraction data. Too much charge density is not accounted for, or over calculated. Similarly, the R value for RbV₆Sb₆ is not great. On the fundamental side, it is unclear if the actual structure is indeed related to AV₃Sb₅ since the claimed crystal structure could be wrong.

Reply: We thank the reviewer for pointing out this issue. To improve the refinement quality, we have collected additional diffraction data of V₆Sb₄. The new data yield smaller R values with $R_{\text{int}} = 0.0719$, $R_1 = 0.0557$ and $wR_2 = 0.1328$. The main peak of the residual charge density (with an intensity of 2.5) locates at the center of the cluster formed by six V atoms in two adjacent layers, where the occupation of an additional atom can cause significant structural instability; this peak should be an artefact arising from the absorption effect of heavy atoms (in our case, the vanadium cluster). For RbV₆Sb₆, we have also recollected the diffraction data and refined the structure based on the new results. The present refinement has smaller R values with $R_{\text{int}} = 0.0868$, $R_1 = 0.0358$ and $wR_2 = 0.0924$. The new refinement results, with much improved quality, give the same crystal structures as we showed in the original manuscript, thus we are confident that the structures are correct. We updated the corresponding refinement parameters in Table S1 of Supplementary Information.

2. In terms of superconductivity, the transport data obtained from 4 probe measurement does not go to zero resistivity at the pressure the authors claim. If fact, the resistivity at the base temperature is likely still numerically larger than a good metal, say copper, at room temperature. Given that a filamentary superconductor would be able to short a sample already, the bulk nature of superconductivity is highly questionable. Possibility of a structural or magnetic phase transition under

high pressure cannot be ruled out.

Reply: First, we would like to emphasize that the zero resistivity is indeed realized in our sample in the pressure range between ~ 30 and ~ 45 GPa (see Fig. R1 for the expanded view of the low-temperature resistance). High pressure often induces cracks in the sample; the resulted domain boundaries can significantly contribute to scattering, thus are in principle detrimental to the zero resistivity (this is likely what happens for the $P > 45$ GPa curves in Fig. R1). Since we observed zero resistance in the superconducting phase (Fig. R1), it is more appropriate to identify it as bulk superconductivity instead of filamentary (considering that the filamentary superconducting paths may not survive the sample cracking under high pressure). The field-dependence of the low-temperature resistance also support the bulk superconductivity scenario: as shown in Supplementary Information Fig. S6, at $P = 35.9$ GPa, the resistive drop with an onset of ~ 1.5 K is gradually suppressed by the applied magnetic fields and is not detectable down to 0.3 K at $B = 1.0$ T. This is fully consistent with the behavior of bulk superconductivity. Such strong field dependence, together with the observation of zero resistance, should exclude the alternative explanations for the resistive drop, e.g., structural or magnetic phase transitions.

Figure R1. Expanded view of the low-temperature resistance of CsV_6Sb_6 at high pressure showing the zero resistance realized at $P = 27.8, 29.7, 33.0, 35.9$ and 39.7 GPa in sample S1. In sample S2, zero resistance is seen at $P = 45.2$ GPa at the lowest temperature.

To check whether there are structural phase transitions at high pressures, we performed the high-pressure X-ray diffraction (XRD) measurements. We found the emergence of superconductivity is related to the occurrence of a structural phase transition under the pressure of 14-22 GPa, above which the original rhombohedral structure distorted to the monoclinic structure. We discuss these results in the revised manuscript and the Supplementary Information.

Reviewer #2 (Remarks to the Author):

In the manuscript entitled «A new class of bilayer kagome lattice compounds with Dirac nodal lines and pressure-induced superconductivity », M. Shi, F. Yu and coworkers report the synthesis of new compounds [AV₆Sb₆, A = K, Rb, Cs, and V₆Sb₄] that host kagome bilayers with linearly dispersing bands forming nodal Dirac lines.

Nowadays, the study of Kagome lattices, in which magnetic frustration, topological ordering, superconductivity and charge ordering are simultaneously present, it represents the frontier in the study of topological materials, being AV₃Sb₅ the main focus of the community. In this respect, the synthesis and study of new compounds sharing similar properties is timely. However, the materials investigated in this manuscript do not exhibit several of the intriguing phenomena that are fascinating the community. As the authors clarify, in fact, those materials have no charge ordering and DFT calculations indicate that they lack magnetic interactions, and a nonmagnetic ground-state is expected.

In general, I found the manuscript well written, and I appreciate the nice description of the symmetry protecting the band degeneracy responsible for the Dirac nodal line. However, in my opinion this is not sufficient to guarantee publication in Nature Communications.

We thank the reviewer for the detailed comments and suggestions. These helped us to improve the manuscript. We are confident that all the concerns and questions can be addressed.

1) The quality of the ARPES data is in fact not sufficient to confirm the theoretical prediction of DFT. The authors themselves are very honest when in the methods they state “the terraced surfaces of the cleaved samples hamper a clear observation of fine electronic features”.

- In particular, we cannot resolve the two states that should be responsible for the Dirac line.

Reply: We thank the reviewer for the suggestion. As we mentioned in the earlier manuscript, ARPES measurements are challenging due to the terraced surfaces of the cleaved samples. However, we agree with the reviewer that improved ARPES data quality is important. Therefore, we have carried out more measurements at 03U beamline of Shanghai Synchrotron Radiation Facility (SSRF) with a small beam spot size. The new results, with improved data quality, are shown in Figure 3 of the revised manuscript. The newly measured band structure, with improved clarity, are largely consistent with the new DFT calculations (Fig. 3h). The two bands responsible for the Dirac line can be resolved now. The band crossing is very close to, or slightly above the Fermi level, which is also consistent with the DFT calculation.

- The assignment of the Fermi level is unclear to me, in fact there are several ARPES signatures that are far from being reproduced by DFT. In Figure 3e, the high intensity at the M point is clearly within the calculated red lines. Similarly, in Figure 3 b the large intensity coincide with the Brillouin zone boundary indicated by yellow lines, and not with the calculations shown by the red dashed line.

Reply: As shown in the revised Figure 3, our new data with improved quality exhibit much better consistency with the new DFT calculations. We also copy the new figure here as Fig. R2, for easing of reading. We thank the reviewer again for the constructive suggestion.

Figure R2. The electronic structure of CsV₆Sb₆. The new ARPES data were obtained at the beamline 03U of the SSRF with photon energy ~60 eV. The calculated bulk band structures of CsV₆Sb₆ at $k_z = 0-0.5$ is obtained from DFT+U calculations with $U = 2$ eV.

2) Given the symmetry of the crystal structure, I would expect multiple surface terminations to be possible, and the authors should characterize their influence on the electronic properties both from the experimental and theoretical point of view.

Reply: This is a good idea. We have followed the reviewer's suggestion and performed a real space scan (shown below as Fig. R3) using the small beam spot (15 μ m*15 μ m) ARPES at 03U beamline of Shanghai Synchrotron Radiation Facility (SSRF). Although we have performed rather careful searching on that, we didn't find multiple sets of band structure originating from different terminations. We note that the absence of multiple sets of band structure is reminiscent of what has been reported on the CsV₃Sb₅ system. This is also consistent with the DFT calculations, in which the Cs orbitals show negligible contribution to the bands near the Fermi level.

Figure R3. ARPES intensity map over the sample surface. The measurement was done on a 40 μm scanning grid and covered a surface area of $\sim 680 \times 680 \mu\text{m}^2$. We didn't find multiple sets of band structures from different terminations.

3) ARPES data might be dominated by surface states, and photon energy dependent study would be necessary to distinguish the surface from bulk contributions.

Reply: We thank the reviewer for the suggestion. It is indeed a good idea to distinguish the surface from bulk contributions with a photon energy dependent study. We have followed the review's suggestion and performed new ARPES measurements with different photon energies (from 42 eV to 60 eV, please see Fig. R4). Nevertheless, the k_z dependence of the bulk bands in this material is very weak, which is also demonstrated in the DFT calculations (Fig. 3g).

Fortunately, the higher quality of the new data allows us to perform a more quantitative comparison with the DFT calculations. The measured band dispersion along the $\bar{\Gamma} - \bar{M}$ direction is compared to the DFT calculations with both bulk and surface bands. The second derivative image as a function of momentum is presented for the ARPES data, in order to highlight the fine features of the bands. As shown in Fig. R5, an overall agreement can be achieved between the measured bands and the calculated bulk bands. Nevertheless, an additional band with relatively weak spectral intensity can also be resolved (as indicated by the dashed lines and arrows in Fig. R5). It does not match any of the bulk bands in DFT calculations (red solid lines). Yet, the location of this feature roughly coincides with the dispersion of the calculated surface band which would intersect the bulk nodal line crossing above E_F . We note that the photoemission intensity of the bulk bands is much stronger than that of the surface bands in this material, and not all the surface bands can be clearly identified due to the relative weak intensity. This dichotomy in photoemission intensity could be related to the ARPES matrix element effect. Nevertheless, both bulk and surface states are indeed observed in the new experiments, as suggested by the reviewer. In the revised version, we have included these data and related discussion in the supplements (Fig. S4 and S5).

Figure R4. ARPES intensity plots along the $\bar{\Gamma} - \bar{M}$ measured with photon energies from 42 to 60 eV at ~ 12 K.

Figure R5. a, The band structures including surface states obtained from DFT calculations. The yellow color denotes the surface energy bands. **b,** The second-order differential photoemission image of a CsV_6Sb_6 single crystal measured along the $\bar{\Gamma} - \bar{M}$ direction. The dashed lines and arrows mark the features corresponding to possible surface states. Solid lines represent the bulk bands.

4) From the theoretical point of view, I do not fully understand Figure 2 d. Parity is evaluated only at 4 TRIM points, while I would have expected 8 contributions (if symmetry consideration lowers the number of points, it should be explained). Since the materials have not a direct gap, the authors should indicate in Panel a and b, the lines where the parities product has been calculated.

Reply: We are grateful to the referee for these valuable comments and questions. As pointed out by the reviewer, the parity arguments for identifying the \mathbb{Z}_2 topological

index are based on products of parity eigenvalues at the eight TRIM points. For a nonmagnetic compound crystalizing in the $R\bar{3}m$ space group (No. 166), the three F (and the three L) points are equivalent, and thus we only need to calculate the parity eigenvalues at the four TRIM points (i.e., Γ , Z, F, and L points). This is similar to the case in other compounds (such as Bi_2Se_3) with the space group $R\bar{3}m$ [e.g., *Nature Phys* **5**, 438 (2009)]. In addition, although these materials are metallic, they possess both time-reversal and inversion symmetry as well as a continuous and symmetry-enforced band-gap near the Fermi level, allowing us to calculate the \mathbb{Z}_2 topological invariant between each pair of bands at the TRIM points by simply analyzing the parity of the wave function. Following the referee's suggestions, we have updated the corresponding statements in the revised manuscript and added information of parities for the high-symmetry momenta in Fig. 2a and b.

In figure 2 d, why the parities of only few selected bands are shown?

Reply: In Fig. 2d we focus on the band topology between adjacent bands near the Fermi level; thus, only the products of parity eigenvalues of three bands (which are within ± 0.5 eV from the Fermi level) are illustrated. In the revised version, we also provide the parity eigenvalues of more bands in the Supplementary Information (see Table S3).

In figure 2 a and 2 b, what do the blue and yellow dashed areas indicate?

Reply: In Figures 2a and 2b, the blue and yellow shaded areas highlight the band-gaps (between the topologically nontrivial bands) near the Fermi level that are presented at the TRIM points. These band gaps allow to using parity eigenvalues to calculate the \mathbb{Z}_2 topological invariant. We added the corresponding information to the caption.

5) The report of a superconducting state under pressure is certainly of interest. However, without a proper characterization of the crystal structure/symmetry is impossible to be sure that this is a property of the crystals described in Figure 1-3. As the authors carefully explained, the Dirac nodal line as well as the Kagome lattices arise from the space group and the lattice symmetry. The application of pressure might be responsible for a structural phase transitions, and this is a central aspect to be verified.

Reply: We thank the reviewer for bringing up this important issue. To track the evolution of crystal structure under high pressure, we performed the high-pressure XRD measurements. We positively identify that the onset pressure of the superconductivity in CsV_6Sb_6 (21.2 GPa) roughly coincides with the stabilization of a pressure-induced monoclinic structure at ~ 22 GPa. The structure transition from rhombohedral to monoclinic occurs under the pressure of ~ 14 -22 GPa (Supplementary Information Fig. S9). The high-pressure monoclinic phase can be considered as the distorted rhombohedral structure in the kagome slabs. Due to the difficulty of determining the precise atom positions from the high-pressure XRD

measurements with limited angle range and peak broadening at high pressure, we cannot nail down the topological properties of band structures for this high-pressure monoclinic structure at this stage. However, we argue that the pressure-induced superconductivity in these bilayer kagome materials is still quite interesting even with the structure distortion, considering that in numerous topological materials the pressure-induced superconductivity is reported to be accompanied by structural transitions [Ref. 37,39; X. Li, et al., PNAS 116, 17696-17700 (2019); X. Li, et al., PNAS 116, 9935-9940 (2018); C. Pei, et al., npj Quantum Materials 6:98 (2021)]. We discuss these results in the revised manuscript and the Supplementary Information.

6) Finally, although novelty is not the reason why I cannot recommend the publication of this manuscript in Nature Communication, I would like to bring to the attention of the authors a manuscript, recently published in Chinese Physical Letters 38 127401 [Structures and Physical Properties of V-Based Kagome Metals CsV₆Sb₆ and CsV₈Sb₁₂], in which the synthesis of AV₆Sb₆ was reported. Hence, I would avoid the use of the term “discovery” in the abstract.

Reply: We appreciate that the review did not take novelty as the weak point of our work, but we do not agree that we should avoid the term “discovery”. Indeed, we posted our manuscript on arXiv (<https://arxiv.org/abs/2110.09782>) on October 19, 2021. This is the first report of the AV₆Sb₆ (A = K, Rb, Cs) series of materials. Q. Yin, et al. posted their work on arXiv on October 21 and submitted it to Chin. Phys. Lett. on October 22, 2021. The corresponding author of the latter work even referred to our data and acknowledged our credit in discovering the AV₆Sb₆ series at two online meetings: NCSC 2021 (Oct. 20) and the CPS Fall Meeting 2021 (Oct. 23).

7) Minor comments:

- The data of Figure 3 b and c are symmetrized. The authors should clearly state this in the caption. If possible add the raw data to the supplementary information.

Reply: We carried out new ARPES measurements with improved data quality as we mentioned above. In the new Figure 3, the Fermi surface images are not symmetrized.

- In few points of the manuscript the authors indicate the References as “refs N,M”. I do not know if that is an error or it is done by purpose.

Reply: We thank the reviewer for this comment. We have corrected the citation format in the revised manuscript.

- At the end of the first paragraph of page 4 the authors say “along the specific high-symmetry paths”, but it is unclear which paths.

Reply: We have explicitly given the path in the revised manuscript, as follows: On the page 4,“along the specific high-symmetry paths (Figs 2a and b)”.....has been changed to“along the specific high-symmetry paths, Γ -Z-F- Γ -L-Z-P, (Figs 2a and b)”

- In general, I do not understand why the Authors stress the possibility of using one single generic stoichiometric formula to describe very different materials, since the authors themselves have nicely discussed how symmetry is the key factor, and not stoichiometry.

Reply: We believe that, despite the AV_6Sb_6 and AV_3Sb_5 compounds have different structural symmetries and subsequently different electronic structures and orders, the application of a generic stoichiometric formula can be potentially inspiring from the materials perspective. The differences between the members in the generic series particularly emphasize the significance of structural modification and, thereby, help us understand the enriched physics in the kagome compounds.

- In general DFT is not a good level of approximation to discuss energy band-gap of few meV. I think that besides better ARPES data, the manuscript would profit of refined calculations (hybrid functional for example) to properly check the band dispersion of the states forming the band-gap.

Reply: We thank the referee for this suggestion. We have further checked the band structures and band gap using different functionals, such as PBE (Perdew-Burke-Ernzerhof), PBE-solid (PBEsol), and the full-potential linearized augmented plane-wave (LAPW) in the Wien2K package (see Fig. R6). The results demonstrate that the Dirac nodal lines near the Fermi level are robust against the selection of functionals. The small size of the band gap is due to the rather weak SOC strength in these vanadium bilayer kagome compounds. This is similar to other kagome A-V-Sb materials, such as the AV_3Sb_5 family, whose DFT band structures match the ARPES data quite well [Phys. Rev. Lett. 125, 247002 (2020)].

Figure R6. A comparison of the band structure in CsV_6Sb_6 calculated using PBE (left), PBE-solid (middle) and LAPW (right). All the functionals yield similar band structures comparing to the DFT calculation results shown in our main text Fig. 2. In particular, the features of the nodal lines are confirmed to be robust against functional selection.

Reviewer #3 (Remarks to the Author):

Shi et al reported a new family of Kagome materials that exhibit topological band structure and superconductivity. Recently, Kagome materials like CsV₃Sb₅ attract tremendous attention for unconventional superconductivity, loop current CDW, and topological states. The present work brings a new group of kagome materials to this family, but with enough differences such as the strong interlayer coupling, nodal line states, and superconductivity under pressure. This work has the potential to represent new features of kagome materials. I would consider it positively for Nature Comm if they can provide more solid information to support their conclusions.

We thank the referee for finding our work “has the potential to represent new features of kagome materials” and providing positive evaluations. We also thank the referee for insightful comments and questions, which have helped us to improve our manuscript.

1. The nodal lines and Z₂ topological invariants are highlighted in the paper, which is mainly supported by calculations. If the theory is correct, there should be signatures of topological surface states due to the nodal lines in the bulk. The authors should present direct calculations on the surface band structure and also reveal them in ARPES.

Reply: This is a good idea. We have performed calculations to show the surface band structure, as suggested by the reviewer. Figure R7a shows the band structures including surface states obtained from DFT calculations, and the yellow color denotes the surface energy bands. In order to look for the potential topological surface states by experiments, improved ARPES data quality is needed. Therefore, we have carried out new ARPES measurements at the Shanghai Synchrotron Radiation Facility (SSRF) and obtained new data for CsV₆Sb₆ with improved quality.

The improved ARPES data quality enables us to perform a more quantitative comparison with the DFT calculations (new Fig. 3 and Fig. R7). The measured band dispersion along the $\bar{\Gamma} - \bar{M}$ direction is compared to the DFT calculations with both bulk and surface bands. The second derivative image as a function of momentum is presented for the ARPES data, in order to highlight the fine features of the bands. As shown in Fig. R7, an overall agreement is achieved between the measured bands and the calculated BULK bands (the red curves in Fig. R7b). Nevertheless, an additional band with relatively weak spectral intensity can also be resolved (as indicated by the dashed lines and arrows in Fig. R7). It does not match any of the bulk bands in DFT calculations (red solid lines). Yet, the location of this feature roughly coincides with the dispersion of the calculated surface band which would intersects the bulk nodal line crossing above E_F, providing possible hints for

the topological surface states. We note that the photoemission intensity of the bulk bands is much stronger than that of the surface bands in this material, and not all the surface bands can be clearly identified due to the relative weak intensity. This dichotomy in photoemission intensity could be related to the ARPES matrix element effect.

In the revised version, we have replaced the ARPES data in Fig. 3 and shown the comparison between ARPES results and calculated surface states in Fig. S5.

Figure R7. **a**, The band structures including surface states obtained from DFT calculations. The yellow color denotes the surface energy bands. **b**, The second-order differential photoemission image of a CsV_6Sb_6 single crystal measured along the $\bar{\Gamma} - \bar{M}$ direction. The dashed lines and arrows mark the features corresponding to possible surface states. Solid lines represent the bulk bands.

2. The materials are heavily electron-doped. In contrast, CVS is commonly hole-doped. Can authors elaborate on the material/chemistry origin behind different doping behaviors?

Reply: In the original version of manuscript the ARPES data quality was not very satisfying, and the relatively low resolution caused an (unintentional) incorrect determination of the Fermi level which is 150 meV above the position in DFT calculation. This, and its indication of a high electron doping, were proved to be inaccurate by our later ARPES measurements. The improved ARPES data (as shown in Fig. 3 and Supplementary Information Fig. S4 and S5) explicitly imply that the actual Fermi level in our CsV_6Sb_6 samples is very close to that in the DFT calculation. Hence, the doping level in this material is indeed low, excluding the possibilities for heavy structural disorder or high density of vacancies.

3. The pressure induces superconductivity. Do authors have any conclusions on the nature of superconductivity? Further, the pressure is rather high up to 80 GPa. Is there any structural transition under pressure? Does the material still preserve the kagome lattice or the topology?

Reply: To further investigate the nature of the superconductivity, we performed high-pressure XRD measurements after the submission of our manuscript. The results confirm the occurrence of a structural transition under pressure of ~14 -22 GPa, where the structure changes from rhombohedral to monoclinic. Hence, we can conclude that the superconductivity happens in the high-pressure monoclinic phase which can be considered as a distorted version of the ambient pressure rhombohedral structure: the in-plane lattice parameter a in the original R-3m structure develops into two unequal values a and c in the monoclinic phase, whilst the angle β becomes smaller than 120° (see Supplementary Information Fig. S8 and S9). We added the high-pressure XRD results to the revised manuscript in the Supplementary Information. Unfortunately, the limited angle range of the XRD measurements and significantly broadened peaks at high pressure prevent further determination of precise atom positions, thus we cannot make further comments on the band structure and topological nontriviality in the high-pressure monoclinic phase. These would be the aims of our future work.

REVIEWER COMMENTS

Reviewer #1 (Remarks to the Author):

The authors have properly addressed my previous comments. Therefore, I recommend acceptance of this revised manuscript.

Reviewer #2 (Remarks to the Author):

I thank the Authors for the efforts paid in revising their manuscript. The inclusion of new data and their analysis has improved overall the quality of the text. At the same time, it shows a common attitude, a bad attitude in my opinion, to submit manuscripts with preliminary and partial results, with the idea to repeat the experiments while the manuscript is under revision, only to gain time.

There are still few important points that I invite the Authors to consider.

- 1) The fact that superconductivity emerges only with the structural transition to the monoclinic phase requires clarifying if the new phase is also "kagome-like". The authors should describe how, on the basis of symmetry arguments, the topological aspects, in particular how nodal lines evolve in the new phase. What can we expect for the topological indices?
- 2) In figure 3 h, the calculations make difficult to appreciate the ARPES data. Since the same theoretical band dispersion is shown in panel g, I suggest the authors to remove the red lines from panel h.
- 3) I would include the photon energy scan in the supplementary information, since it provides valuable information (the weak k_z dispersion). I suggest to show also the evolution as a function of photon energy (or k_z) of momentum distribution curves (for example at the Fermi level and at -0.2 eV) and energy distribution curves taken (at Γ and at M).
- 4) Following the previous comment, it would be particularly interesting to highlight in this k_z scan the nodal-line, which according to figure 2c should be accessible in the plane probed by the photon energy scan.
- 5) The feature indicated as surface state in Figure S5 is, in my opinion, only an effect of the second derivative and I refrain the authors from the use of derivative.
- 6) In the reply, the Authors show in Figure R3 an intensity map over the sample surface. But I cannot find a description of where the intensity has been integrated.
- 7) I still think that We, as a community, should avoid the use of terms like "discovery". In any case, the material has been subject of a previous publication and I find dangerous the idea to use the day of submission on arXiv to establish "who comes first". This attitude is what is causing a general decline of the manuscripts quality on arXiv, which are often incomplete, with preliminary results.

Reviewer #3 (Remarks to the Author):

The Authors answered my technical questions. The band crossing (nodal line) points are verified by ARPES. But the significant consequence of nodal line, surface states, are still missing. Superconductivity emerges under pressure. But it is unclear how interesting the superconductivity can potentially become. I would agree with some comments of the 2nd referee, "However, the materials investigated in this manuscript do not exhibit several of the intriguing phenomena that are fascinating the community", albeit these materials indeed show a kagome structure. I hesitate to recommend the publication of this work in Nature Commun..

Reviewer #1 (Remarks to the Author):The authors have properly addressed my previous comments. Therefore, I recommend acceptance of this revised manuscript.

We thank Reviewer#1 for supporting publication in Nature Communications. We are also grateful to the reviewer for his/her comments and questions which helped us to improve the manuscript.

Reviewer #2 (Remarks to the Author):

I thank the Authors for the efforts paid in revising their manuscript. The inclusion of new data and their analysis has improved overall the quality of the text. At the same time, it shows a common attitude, a bad attitude in my opinion, to submit manuscripts with preliminary and partial results, with the idea to repeat the experiments while the manuscript is under revision, only to gain time.

We appreciate that Reviewer#2 confirmed the improved quality of our manuscript.

There are still few important points that I invite the Authors to consider.

1) The fact that superconductivity emerges only with the structural transition to the monoclinic phase requires clarifying if the new phase is also “kagome-like” . The authors should describe how, on the basis of symmetry arguments, the topological aspects, in particular how nodal lines evolve in the new phase. What can we expect for the topological indices?

Reply: We thank Reviewer#2 for bringing up this issue. Unfortunately, due to the limited angle range of our high-pressure XRD measurements and considerably broadened peaks at high pressure, we cannot determine the atom positions precisely. Hence, the present high-pressure XRD data is premature for making further comments on the band structure and topological nontriviality in the high-pressure monoclinic phase. Further detailed investigations under high-pressure condition are required to nail down the topological aspects of this phase; these are beyond the scope of the present work, which is mainly focused on the topological properties of these compounds at the ambient pressure.

2) In figure 3 h, the calculations make difficult to appreciate the ARPES data. Since the same theoretical band dispersion is shown in panel g, I suggest the authors to remove the red lines from panel h.

Reply: We remove the red lines in Fig.3h following Reviewer#2's suggestion.

3) I would include the photon energy scan in the supplementary information, since it provides valuable information (the weak k_z dispersion). I suggest to show also the evolution as a function of photon energy (or k_z) of momentum distribution curves (for example at the Fermi level and at - 0.2 eV) and energy distribution curves taken (at Γ and at M).

Reply: We thank the reviewer for this suggestion. The photon energy scan is already presented in the Supplementary Fig. S4. The momentum distribution curves and energy distribution curves are now shown in Fig. S5 in the Supplementary Information.

4) Following the previous comment, it would be particularly interesting to highlight in this k_z scan the nodal-line, which according to figure 2c should be accessible in the plane probed by the photon energy scan.

Reply: We agree with Reviewer#2 that momentum-wise the nodal lines can be resolved in the k_z scan. Indeed, two straight features along k_z show up in our constant energy contour (Fig. R1), whose locations are consistent with the bands contributing to the Dirac crossings in our DFT calculations (see the white arrows in Fig. R1b). These can be indications of the band crossings. Meanwhile, we would like to point out that the calculated Dirac crossings can be exactly at or slight above the Fermi level, depending on the value of U and k_z (Fig. 2a and Fig. 3g). Accordingly, it is technically difficult to resolve these crossings clearly in the ARPES measurements. Therefore, although our k_z scan is consistent with the existence of the nodal-line, we would like to be more conservative on claiming the experimental observation of the Dirac crossings. We included the data shown in Fig. R1 in the Supplementary Fig. S5.

Figure R1. **a**, Photon-energy dependent ARPES measurements of constant energy contour in k_z - k_x plane at the Fermi level. **b**, an expanded view showing the likely positions of the nodal lines formed by type-II Dirac cones along k_z (indicated by the white arrows).

5) The feature indicated as surface state in Figure S5 is, in my opinion, only an effect of the second derivative and I refrain the authors from the use of derivative.

Reply: We are grateful to the reviewer for pointing out this issue. In the last version of our manuscript, the “second derivative” plot shown in the panel b of Supplementary Fig. S5 was indeed a curvature plot. We apologize for this typo and have corrected the description in the updated manuscript. Here we display both the curvature plot and the second derivative plot in Fig. R2, both of which have been broadly used by the ARPES community to highlight fine features in the photoemission spectra. The weak signatures along the $\bar{\Gamma}$ - \bar{M} direction appear in both plots. Therefore, they are unlikely to be artifacts introduced by either of the above data analysis methods. Again, we thank the reviewer for helping us to find a mistake in our manuscript. We have now corrected the typo and included both the curvature plot and the second derivative plot in the updated Supplementary Fig. S6, which are also shown here in Fig. R2 for the ease of reading.

Figure R2. **a**, The curvature plot and **b**, the second derivative plot of the ARPES data measured in a CsV_6Sb_6 single crystal along the $\bar{\Gamma} - \bar{M}$ direction. The solid lines represent the bulk bands. The dashed lines and arrows mark the features corresponding to possible surface states

6) In the reply, the Authors show in Figure R3 an intensity map over the sample surface. But I cannot find a description of where the intensity has been integrated.

Reply: We show Figure R3 once again in this point-to-point reply. The intensity map was obtained by spatial scanning of 40 μm per step and each spectrum was integrated from 1.3 eV below Fermi level.

Figure R3. ARPES intensity map over the sample surface. The measurement was done on a 40 μm scanning grid and covered a surface area of $\sim 680 \times 680 \mu\text{m}^2$ and each spectrum was integrated from 1.3 eV below Fermi level.

7) I still think that We, as a community, should avoid the use of terms like “discovery” . In any case, the material has been subject of a previous publication and I find dangerous the idea to use the day of submission on arXiv to establish “who comes first” . This attitude is what is causing a general decline of the manuscripts quality on arXiv, which are often incomplete, with preliminary results.

Reply: We changed the phrases correspondingly in the updated manuscript.

Reviewer #3 (Remarks to the Author):

The Authors answered my technical questions. The band crossing (nodal line) points are verified by ARPES. But the significant consequence of nodal line, surface states, are still missing. Superconductivity emerges under pressure. But it is unclear how interesting the superconductivity can potentially become. I would agree with some comments of the 2nd referee, “However, the materials investigated in this manuscript do not exhibit several of the intriguing phenomena that are fascinating the

community” , albeit these materials indeed show a kagome structure. I hesitate to recommend the publication of this work in Nature Commun..

We thank the Reviewer#3 for the careful review and valuable comments. Here, we would like to emphasize that our work represents the very first report of the topological nontriviality and pressure-induced superconductivity in a new class of double-layer kagome lattice materials. In this sense, it definitely opens an unprecedented route for investigating the highly enriched physical phenomena in the family of materials with kagome structure. It is well known that the electronic ground states and their properties in kagome materials are highly sensitive to many factors, including elements comprising the kagome nets and spacing layers, the detailed structure of kagome networks, band fillings, etc. Our report can promisingly make provision for further discovery of fascinating phenomena in this family of compounds via the manipulation of these factors (by the means of, e.g., external pressure, uniaxial stress or chemical doping). Moreover, whilst some electronic orderings and symmetry-breaking phenomena that has been observed in the single-layer AV_3Sb_5 compounds are missing here, the difference between these two families (in particular the distinct topological identifications of the single-layer and bilayers) undoubtedly help us to better understand the complicated physics in the kagome compounds. Last but not least, from the materials perspective, our work potentially provides inspirations for future realization of other multi-layer kagome lattice compounds (which are very rare currently). With these notes, we believe that our work would be interesting to a broad audience of the physics community and thus respectfully disagree with the Reviewer’s judgment on the significance of our work.

REVIEWER COMMENTS

Reviewer #2 (Remarks to the Author):

I thank the Authors for their detailed answers and for having included new information in the supplementary materials. The study of topological kagome metals is a very hot field of research, and I agree with the Authors that the discovery of new compounds, containing bilayer of kagome units, may attract considerable attention and motivate further studies, for example to elucidate the origin of the superconducting phase under pressure, and its topological properties. To this extent, this new class of compounds may still represent an interesting opportunity in the field, although it lacks some interesting physical effects, such as charge orderings, that characterize the XV_3Sb_5 family.

However, there is still one point in the present manuscript that does not fully convince me. In my previous comment 5, I was not referring to the specific use of second derivative, but more in general to all the treatments (including curvature methods) used to alter the data. In my opinion, there is no trace of a surface state, neither in the original data nor in the curvature. The fact that curvature method is used by a fraction of the ARPES community it does not legitimate its use, which should be discouraged. Derivative and curvature methods can guide the eye of readers, but the presence of the band should be clearly confirmed in the original spectra. If the authors want to confirm the existence of a surface state in this new compound, they should show the corresponding peak dispersion in the EDCs. Otherwise I suggest the authors to remove the discussion of the surface state, leaving it for future study similarly to the characterization of the SC and its topology.

Reviewer #2 (Remarks to the Author):

I thank the Authors for their detailed answers and for having included new information in the supplementary materials. The study of topological kagome metals is a very hot field of research, and I agree with the Authors that the discovery of new compounds, containing bilayer of kagome units, may attract considerable attention and motivate further studies, for example to elucidate the origin of the superconducting phase under pressure, and its topological properties. To this extent, this new class of compounds may still represent an interesting opportunity in the field, although it lacks some interesting physical effects, such as charge orderings, that characterize the XV_3Sb_5 family.

However, there is still one point in the present manuscript that does not fully convince me. In my previous comment 5, I was not referring to the specific use of second derivative, but more in general to all the treatments (including curvature methods) used to alter the data. In my opinion, there is no trace of a surface state, neither in the original data nor in the curvature. The fact that curvature method is used by a fraction of the ARPES community it does not legitimate its use, which should be discouraged. Derivative and curvature methods can guide the eye of readers, but the presence of the band should be clearly confirmed in the original spectra. If the authors want to confirm the existence of a surface state in this new compound, they should show the corresponding peak dispersion in the EDCs. Otherwise I suggest the authors to remove the discussion of the surface state, leaving it for future study similarly to the characterization of the SC and its topology.

Reply: We appreciate that Reviewer#2 pointed out the novelty of our paper. We agree with reviewer's comment that the derivative/curvature analysis is not enough to confirm the existence of topological surface states; we removed the discussion of the surface state (Fig. S6 and related descriptions in the main text) following such suggestion. We are grateful to the reviewer for his/her comments and questions which helped us to improve the manuscript.